# Model Agnostic Multilevel Explanations

Karthikeyan Natesan Ramamurthy, Bhanukiran Vinzamuri, Yunfeng Zhang, Amit Dhurandhar

IBM Research, Yorktown Heights, NY USA 10598
knatesa@us.ibm.com, bhanu.vinzamuri@ibm.com, {zhangyun, adhuran}@us.ibm.com

## Abstract

In recent years, post-hoc local instance-level and global dataset-level explainability of black-box models has received a lot of attention. Lesser attention has been given to obtaining insights at intermediate or group levels, which is a need outlined in recent works that study the challenges in realizing the guidelines in the General Data Protection Regulation (GDPR). In this paper, we propose a meta-method that, given a typical local explainability method, can build a multilevel explanation tree. The leaves of this tree correspond to local explanations, the root corresponds to global explanation, and intermediate levels correspond to explanations for groups of data points that it automatically clusters. The method can also leverage side information, where users can specify points for which they may want the explanations to be similar. We argue that such a multilevel structure can also be an effective form of communication, where one could obtain few explanations that characterize the entire dataset by considering an appropriate level in our explanation tree. Explanations for novel test points can be cost-efficiently obtained by associating them with the closest training points. When the local explainability technique is generalized additive (viz. LIME, GAMs), we develop fast approximate algorithm for building the multilevel tree and study its convergence behavior. We show that we produce high fidelity sparse explanations on several public datasets and also validate the effectiveness of the proposed technique based on two human studies – one with experts and the other with non-expert users – on real world datasets.

## 1 Introduction

A very natural and effective way to communicate is to first provide high level general concepts and then only dive into more of the specifics [1]. In addition, the transition from high level concepts to more and more specific explanations should ideally be as logical or smooth as possible [2, 3]. For example, when you call a service provider there is usually an automated message trying categorize the problem at a high level followed by more specific questions. Eventually if the issue is not resolved a human representative may intervene to delve into further details. In such cases, information or explanations you provide at multiple levels enables others to obtain insights that are otherwise opaque. Recent work [4] has stressed the importance of having such multilevel explanations to successfully meet the requirements of Europe's General Data Protection Regulation (GDPR) [5]. They argue that simply having local or global explanations may not suffice for providing satisfactory explanations in many cases. In fact, even in the widely participated FICO explainability challenge [6] it was expected that one provides not just local explanations but also insights at the intermediate class level.

Motivated by this need, in this paper, we propose a novel model agnostic multilevel explanation (MAME) method that takes as input a post-hoc local explainability technique for black-box models (*e.g.* LIME [7]) and an unlabeled dataset. The method then generates multiple explanations for each of the examples corresponding to different degrees of cohesion (i.e. parameter tying) between explanations of the examples. This explicitly *controllable* degree of cohesion determines a level in our multilevel explanation tree. In addition, we constrain that the predictions of the explanation model to be close to that of the black-box at each tree node, ensuring fidelity. At the extremes, the

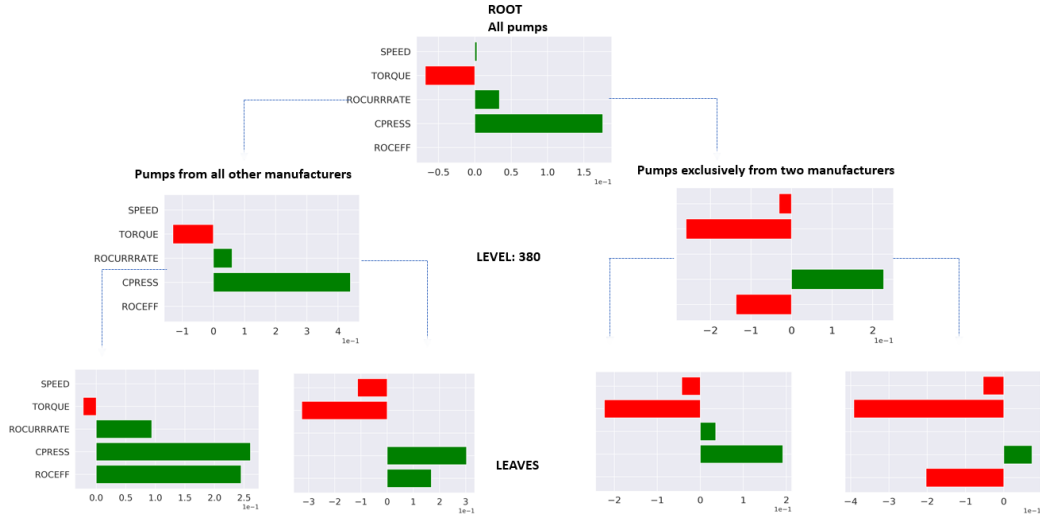

Figure 1: Illustration of multilevel explanations generated by MAME for an industrial pump failure dataset consisting of 2500 wells. We show three levels: the bottom level (four) leaves which correspond to example local explanations, the top level corresponds to one global explanation and an intermediate level corresponds to explanations for two groups highlighted by MAME. Based on expert feedback, these intermediate explanations, although explaining the same type of pump, had semantic meaning as they corresponded to *different manufacturer groups* that behave noticeably differently.

leaves would correspond to independent local explanations as with methods like LIME, while the root would correspond to a single global explanation given the high degree of cohesion.

An illustration of this is given in Figure 1, where multilevel explanations were generated by MAME for a real industrial pump failure dataset (see Section 4.4 for details). We show three levels: the four leaves correspond to example local explanations (amongst many), the root corresponds to one global explanation and an intermediate level corresponds to explanations for two groups highlighted by MAME. Note that levels are numbered from 1 (leaves) increasing up to the highest value (root). The dotted lines indicate that the nodes are descendants of the node above, but not direct children. Based on expert feedback these intermediate explanations correspond to pumps having different manufacturers resulting in noticeable difference in behaviors. Also note that each level provides distinct enough information not subsumed by just local or global explanations, thus motivating the need for such multilevel explanations. Such explanations can thus help identify key characteristics that bind together different examples at various levels of granularity. They can also provide *exemplar based explanations* based on the groupings at specific levels. These are provided in the supplement.

Our method can also take into account side information such as similarity in explanations based on class labels or user specified groupings based on domain knowledge for a subset of examples. Moreover, one can also use non-linear additive models going beyond LIME to generate local explanations. Our method thus provides a lot of *flexibility* in building multilevel explanations that can be customized apropos a specific application. We prove that our method actually *forms a tree* in that examples merged in a particular level of the tree remain together at higher levels. The proposed *fast approximate algorithm* for obtaining multilevel explanations is proved to converge to the exact solution. We show that we produce high fidelity sparse explanations on several public datasets. We also validate the effectiveness of the proposed technique based on two human studies – one with experts and the other with non-expert users – on real world datasets.

## 2   Related Work

The most traditional direction in explainable AI is to directly build interpretable models such as rule lists or decision sets [8, 9] on the original data itself so that no post-hoc interpretability methods are required to uncover the logic behind the proposed actions. These methods however, may not readily

give the best performance in many applications compared to more sophisticated black-box models. Some works try to provide local [7, 10, 11, 12] as well as global explanations that are both feature and exemplar based [13]. The method proposed by Plumb *et al.* [13] provides only local explanations and detects global patterns, but does not automatically identify and provide explanations for groups of data. Exemplar based explanations [14, 15] identify few examples that are representative of a larger dataset. Previous works tend to use distinct models to provide the local and global explanations. Hence, consistency between these models could potentially be an issue. Pedreschi *et al.* [16] propose a high-level approach to cluster local rule-based explanations to learn a local to global tree structure. The authors do not explicitly provide any algorithm and the drawback here is that the explanations at levels other than local may not have high fidelity to the black-box. In TreeExplainer [17] the authors present five new methods to combine efficiently computed exact local SHAP explanations which also additionally account for local feature interactions to understand global behavior of the model. Drawbacks are similar to [16]. Tsang *et al.* [18] propose a hierarchical method to study the change in the behavior of interactions for a local explanation from an instance level to across the whole dataset. A key difference is that they do not fuse explanations as we do as they go higher up the tree. Moreover, their notion of hierarchy is based on learning higher order interactions between the prediction and input features, which is different from ours. Bhatt *et al.* [19] propose a method to aggregate explanation functions based on various desiderata such as complexity, faithfulness, etc. Lakkaraju *et al.* [20] present a model agnostic approach to customize local explanations to an end-user defined feature set of interest. In contrast, MAME allows the end user to specify which explanations are similar apriori and also controls for complexity (sparsity) to learn high fidelity explanations.

Our approach has relations to convex clustering [21, 22], and its generalizations to multi-task learning [23, 24]. However, our goal is completely different (multilevel post-hoc explainability) and our methodology of computing and using local models that mimic black-box predictions is also different.

## 3   Method

Let $X \times Y$ denote the input-output space and $f : X \to Y$ be a classification/regression function corresponding to a black-box model. Let $g(.)$ be a potentially non-linear map on feature vector $x \in X$. Let $p$ denote the number of features and $n$ the number of instances. For the parameter vector $\theta \in \mathbb{R}^p$, $l(x, \theta) = g(x)^T \theta$ is a generalized additive model, $g(.)$ being a pre-specified map. We can learn this from the predictions of $f(.)$ for examples near $x$. This provides a local explanation for $x$ given by $\theta$. The similarity, $\psi(x, z)$, between $x$ and $z$, can be estimated as $\exp(-d(x, z)/\eta)$, $d(., .)$ being the distance function, and $\eta$ being the bandwidth. Let $\{(x_1, y_1), \ldots, (x_n, y_n)\}$ be a dataset of size $n$, where $y_i$ may or may not be known for each $i$. For each $x_i$, we define the neighborhood $\mathcal{N}_i = \{z \in X | \psi(x_i, z) \geq \kappa\}$, $\kappa$ close to 1. In practice, $\mathcal{N}_i$ of size $m$ can be generated by randomly perturbing the instance $(x_i)$ $m$ number of times as done in previous works [7] . We now define the optimization:

$$\min_{\Theta^{(\beta)}} \sum_{i=1}^{n} \sum_{z \in \mathcal{N}_i} \psi\left(x_i, z\right) \left(f(z) - g(z)^T \theta_i\right)^2 + \alpha_i ||\theta_i||_1 + \beta \left(\sum_{i<j} w_{ij} ||\theta_i - \theta_j||_2\right) \qquad (1)$$

where $\alpha_1, ..., \alpha_n$, $\beta \geq 0$ are regularization parameters, $w_{ij} \geq 0$ are custom weights and $\Theta^{(\beta)}$ is a set of $\theta_i$ $\forall i \in \{1, ..., n\}$ for a given $\beta$.

The first term in (1) tries to make the local models for each example to be as faithful as possible to the black-box model, in the neighborhood of the example. The second term tries to keep each explanation $\theta_i$ sparse. The third term tries to group together explanations. This in conjunction with the first term has the effect of trying to make *explanations of similar examples to be similar*. Here we have the opportunity to *inject domain knowledge* by creating a prior knowledge graph with adjacency matrix $W$. The edge weights $w_{ij}$ can be set to high values for pairs of examples that we consider to have similar explanations, while setting zero weights for the rest. In the third term, other norms instead of $\ell_2$ such as $\ell_1$ or $\ell_\infty$ can be used as well, but we find that $\ell_2$ provides strong theoretical guarantees and good experimental results.

We solve the above objective for different values of $\beta$, wherein $\beta = 0$ corresponds to the leaves of the multilevel explanation tree. Each leaf represents a training example and its local LIME explanation. At $\beta = 0$, (1) decouples to $n$ optimizations, corresponding to the LIME explanations. $\beta$ can be adaptively increased from 0 resulting in progressive grouping of explanations (since $\theta$s get closer),

and their corresponding examples, forming higher levels of the tree. The grouping happens because $\theta_i$ and $\theta_j$ with a non-zero $w_{ij}$ are encouraged to get closer as $\beta$ increases in (1). The intermediate levels hence correspond to disjoint clusters of examples with their representative explanations. The root of the tree obtained at a high $\beta$ value represents the global explanation for the entire dataset. Note that the number of unique explanations as $\beta \to \infty$ is equal to the number of connected components in the prior knowledge graph. Throughout this paper, we assume that this graph has a single connected component, without loss of generality.

## 3.1 Optimization Details

We solve the optimization in (1) using ADMM [25] by posing (1) as

$$
\min_{\Theta^{(\beta)}, U^{(\beta)}, V^{(\beta)}} \sum_{i=1}^{n} \sum_{z \in \mathcal{N}_i} \psi(x_i, z) \left( f(z) - g(z)^T \theta_i \right)^2 + \alpha_i \|u_i\|_1 + \beta \left( \sum_{e_l \in \mathcal{E}} w_l \|v_l\|_2 \right),
$$
$$
\text{such that } \Theta = U, \ \Theta D = V. \tag{2}
$$

The augmented Lagrangian with scaled dual variables is

$$
\min_{\Theta, U, V, Z_1, Z_2} \sum_{i=1}^{n} \sum_{z \in \mathcal{N}_i} \psi(x_i, z) \left( f(z) - g(z)^T \theta_i \right)^2 + \alpha_i \|u_i\|_1 + \beta \left( \sum_{e_l \in \mathcal{E}} w_l \|v_l\|_2 \right) +
$$
$$
\frac{\rho}{2} \|\Theta - U + Z_1\|_F^2 + \frac{\rho}{2} \|\Theta D - V + Z_2\|_F^2. \tag{3}
$$

Here, $U \in \mathbb{R}^{p \times n}$, and $V \in \mathbb{R}^{p \times |\mathcal{E}|}$ are the auxiliary variables, and $\mathcal{E}$ is the list of edges in the prior knowledge graph with non-zero weights. The columns of $U$ and $V$ are denoted by $u_i$ and $v_l$ respectively. $u_i$ corresponds to the same column in $\Theta$ ($\theta_i$), and $D \in \mathbb{R}^{n \times |\mathcal{E}|}$ acts on $\Theta$ to encode differences in their columns. For example, the column of $D$ that encodes $\theta_i - \theta_j$ will contain 1 at row $i$ and $-1$ at row $j$. $Z_1$ and $Z_2$ are the scaled dual variables. This reformulation is inspired by [22].

The ADMM iterations for a given value of $\beta$ are:

$$
\Theta^{(k+1)} = \arg\min_{\Theta} \sum_{i=1}^{n} \sum_{z \in \mathcal{N}_i} \psi(x_i, z) \left( f(z) - g(z)^T \theta_i^{(k)} \right)^2 + \frac{\rho}{2} \|\Theta^{(k)} - U^{(k)} + Z_1^{(k)}\|_F^2 +
$$
$$
\frac{\rho}{2} \|\Theta^{(k)} D - V^{(k)} + Z_2^{(k)}\|_F^2, \tag{4}
$$
$$
U^{(k+1)} = \arg\min_{U} \sum_i \alpha_i \|u_i^{(k)}\|_1 + \frac{\rho}{2} \|\Theta^{(k+1)} - U^{(k)} + Z_1^{(k)}\|_F^2, \tag{5}
$$
$$
V^{(k+1)} = \arg\min_{V} \beta \left( \sum_{e_l \in \mathcal{E}} w_l \|v_l^{(k)}\|_2 \right) + \frac{\rho}{2} \|\Theta^{(k+1)} D - V^{(k)} + Z_2^{(k)}\|_F^2, \tag{6}
$$
$$
Z_1^{(k+1)} = Z_1^{(k)} + \Theta^{(k+1)} - U^{(k+1)}, \tag{7}
$$
$$
Z_2^{(k+1)} = Z_2^{(k)} + \Theta^{(k+1)} D - V^{(k+1)}. \tag{8}
$$

Since (4)-(8) should be solved for progressively increasing values of $\beta$, we adopt the idea of Algorithmic Regularization (AR) [22] to run (4)-(8) only once for each value of $\beta$. The algorithm *regularizes* itself (hence the name) by warm-starting the next set of ADMM iterations with the estimate for the previous $\beta$ value. This method exploits the fact that the solutions for two close $\beta$ values should be close. The $\beta$ values are obtained by initializing to a small $\epsilon$ and multiplying it by a step size $t(> 1.0)$ for the next $k$. We will denote these approximate solutions as $\Theta^{(k)}$, where $k$ corresponds to the index of the set of $\beta$ values. The detailed algorithm for obtaining the multilevel tree using MAME is described in Algorithm 1. This procedure results in a single tree because a single Union operation merges two child nodes (and the sub-trees for which they are root) to create a parent node. Merges only happen above and there will be no cycles hence. In this algorithm, when the stopping criteria on $V$ is met, a single global model for the entire dataset is obtained (represented by the root of the tree). Step (v) creates runs LASSO on the group of examples in each node to get post-processed representative explanations. Since the algorithm implements the AR approach, $\gamma^{(k)}$ is

---

**Algorithm 1** Model Agnostic Multilevel Explanation (MAME) method

---

**Input:** Dataset $x_1, ..., x_n$, black-box model $f(.)$, the coordinate wise map $g(.)$ and prior knowledge graph adjacency matrix $W$.

i) Sample neighborhoods $\mathcal{N}_i$ for each example $x_i$.

ii) Construct matrix $D$ based on edge list $\mathcal{E}$.

iii) Initialize $\Theta^{(0)}, U^{(0)}, V^{(0)}$ and set $k = 0$, multiplicative step-size $t$ (say to $1.01$), $\gamma^{(0)} = \epsilon$ (say to $1e - 10$), grouping threshold $\tau$ (say to $1e - 6$), $\rho$ (say to 2), and $tol.$ (say to $1e - 6$).

iv) Initialize a disjoint set with leaves of the multilevel the tree $S = \{1, ..., n\}$.

**while** $\|V^{k+1}\| > tol.$ **do**

    Obtain $\Theta^{(k+1)}$ by solving (4), $U^{(k+1)}$ by solving (5), $V^{(k+1)}$ by solving (6) with $\beta$ set as $\gamma^{(k)}$.

    Obtain $Z_1^{(k+1)}, Z_2^{(k+1)}$ by solving (7) and (8).

    For each edge $e_l = (i, j) \in \mathcal{E}$, if $\|v_l\| < \tau$, perform Union$(i, j)$.

    $k = k + 1$; $\gamma^{(k+1)} = \gamma^{(k)} * t$.

**end while**

iv) Recover the multilevel tree by keeping track of the disjoint set unions.

v) For every group of examples in each tree node, post-process to get representative explanations by optimizing (1) with $\beta = 0$ only for the examples in the group.

---

used as a surrogate notation for $\beta$ (which is used in the exact solution). The dominant complexity for pass of the *while* loop in Algorithm 1 is $O(pn(p + n))$. Detailed computational complexity analysis is provided in the supplement.

The exact solution where (4)-(8) are run until convergence for each $\beta$ value, will be denoted as $\Theta^{(\beta)}$. We show that the approximate solution converges to the exact one (proof in supplement). This theorem holds true for $\{\Theta^{(k)}\}$ without the post-processing step (v) in Algorithm 1.

**Theorem 3.1.** *As $(t, \epsilon) \to (1, 0)$, where $t$ is the multiplicative step-size update and $\epsilon$ is the initial regularization level, the sequence of AR-based primal solutions $\{\Theta^{(k)}\}$, and the sequence of exact primal solutions $\{\Theta^{(\beta)}\}$ converge in the following sense.*

$$\max\left\{ E_\beta\left(\inf_k \|\Theta^{(k)} - \Theta^{(\beta)}\|\right), E_k\left(\inf_\beta \|\Theta^{(k)} - \Theta^{(\beta)}\|\right) \right\} \xrightarrow{(t,\epsilon)\to(1,0)} 0 \qquad (9)$$

Comparisons of approximation quality and timing between the two solutions are in supplement. We now show that our method actually forms a tree in that explanations of examples that are close together at lower levels will remain at least equally close at higher levels (proof in supplement).

**Lemma 3.2** (Non-expansive map for exact solutions). *If $\beta_1, ..., \beta_k$ are regularization parameters for the last term in (1) for $r$ consecutive levels in our multilevel explanations where $\beta_1 = 0$ is the lowest level with $\theta_{i,s}$ and $\theta_{j,s}$ denoting the (globally) optimal coefficient vectors (or explanations) for $x_i$ and $x_j$ respectively corresponding to level $s \in \{1, ..., r\}$, then for $s > 1$ and $w_{ij} > 0$ we have $\|\theta_{i,s} - \theta_{j,s}\|_2 \leq \|\theta_{i,s-1} - \theta_{j,s-1}\|_2$.*

## 4 Experiments

We evaluate our method on three different cases with two baselines (described below): (a) Two Step, and (b) Submodular Pick LIME (SP-LIME) [7]. We set $g(.)$ to be an identity map for numerical features and one-hot encoding for categorical features in all the experiments for ease of demonstration, though it can be set to any appropriate non-linear map by the user. We first show quantitative benefit of our method in terms of two measures defined in Section 4.2 on several public datasets. The MAME approach has better explanation fidelity, and the explanations are better matched to the important features of the black-box, when compared to the baselines. Secondly, we conducted a study with data scientist users who were not experts in finance using a public loan approval dataset. We found that our method was significantly better insights to these non-experts compared to the baselines. Data scientists are the right catcher for our method as a recent study [26] claims that data scientists are the first consumers of explanations from AI based in most organizations. Finally, we performed a case study involving human experts in the Oil & Gas industry. Insights provided by MAME were semantically meaningful to the experts both when they did and did not provide additional side information.

## 4.1 Baselines and Miscellaneous Details

**Two Step:** This is a hierarchical convex clustering [21, 22] of the local LIME explanations for $n$ instances, given by $\Omega \in \mathbb{R}^{p \times n}$. The objective is written as $\min_{\tilde{\Theta}} \|\Omega - \tilde{\Theta}\|_F^2 + \beta \left( \sum_{i<j} w_{ij} \|\tilde{\theta}_i - \tilde{\theta}_j\|_2 \right)$, where $\|.\|_F$ denotes the Frobenius norm, and $\tilde{\Theta}$ is the Two Step solution, differentiated from the MAME solution given by $\Theta$. As $\beta$ varies, Two Step also results in a multilevel tree with disjoint data clusters at each level of the tree like MAME, although it does not explicitly ensure fidelity to the black-box model predictions, like MAME does (see (1)). We implement Two Step also using an AR scheme [22]. The post-processed explanations for a node in the Two Step tree is the median of the group of explanations in that node.

**SP-LIME:** The submodular pick algorithm [7] chooses a subset of diverse, representative set of explanations from the local LIME explanations. We vary the size of this representative subset from $1$ to $n$. Each representative explanation is associated to a representative training example.

**Explanation for a novel test example:** For Two Step and MAME, the explanation for a novel test example at any level chosen to be the explanation for the nearest training example (based on euclidean distance). For SP-LIME, we identify the nearest representative training example to the test example, and choose its corresponding explanation. The $1-$nearest neighbor association is meaningful since LIME explanations (which also form the basis for Two Step and MAME) are optimized to work well in the neighborhood of each training data point.

## 4.2 Quantitative Evaluation with Public Datasets

**Quantitative Metrics:** We wish to primarily answer two questions quantitatively: (i) How trustworthy the learned local explanation models are at different levels in the tree? (ii) How faithful are the explanation models to the black-box that they explain? We let $n$ be the number of training samples, $l$ be the number of levels, $\theta_i^{(k)}$ be the explanation of sample $i$ at level $k$, and $\theta_{ij}^{(k)}$ be its $j^{\text{th}}$ coefficient.

a) *Explanation Infidelity:* The trustworthiness of the explanations is measured using the infidelity measure proposed in [27] with *novel test data* different than the training data. This measure captures how well our explanation can track changes in black-box prediction when the input goes from a chosen baseline to the actual value. It is defined as $((x - x_0)^T \theta(x) - (f(x) - f(x_0)))^2$ where $x$ is a test example, $\theta(x)$ is the explanation corresponding to it, and $x_0$ is the baseline perturbation. We choose $x_0$ to be $0$ for numerical values and the least frequent category for categorical values. This measure quantifies the fidelity of the explanation to the black-box predictor under perturbations (lower the better). We average this over all test examples and all levels of the tree for Two Step and MAME. For SP-LIME, this measure is averaged over all test examples, and all possible number of representative explanations ranging from 1 to $n$. This averaging ensures that the comparisons are fair. Details of another metric, *generalized fidelity*, used to gauge trustworthiness are in the supplement. Both explanation infidelity and generalized fidelity measures can be situated under the definition of faithfulness given in [28, Corr. 1.2].

b) *Feature importance correlation:* We compute the feature importances of coefficients with LIME, Two Step and MAME methods. For LIME, the importance score of a feature $j$ is defined as $\sqrt{\sum_{i=1}^n |\theta_{ij}|}$ [7]. For Two Step and MAME, we include all the levels to define this as $\sqrt{\sum_{i=1}^n \sum_{k=1}^l |\theta_{ij}^{(k)}|}$. We then compute the Pearson correlation coefficient between these and the black-box model feature importances. Note that this comparison can be performed only for black-box models that can output feature importance scores.

**Setup and Results:** Our demonstration includes: *Auto MPG* [29], *Retention*[1] [30], *Home Line Equity Line of Credit (HELOC)* [6], *Waveform* [29], and *Airline Travel Information System (ATIS)* datasets. The Auto MPG dataset has a continuous outcome variable whereas the rest are discrete outcomes. The (number of examples, number of features) in the datasets are: Auto MPG (392, 7), Retention (1200, 8), HELOC (1000, 22), Waveform (5000, 21), and ATIS (5871, 128). ATIS is a complex, high dimensional, featurized NLP data widely used for intent classification of text (26 classes) from airline customers. We perform $5-$fold cross validation for all datasets except ATIS (which comes with its

| Dataset | Random Forest | | | MLP | | | Random Forest | | |
|---|---|---|---|---|---|---|---|---|---|
| | SP-LIME | Two Step | MAME | SP-LIME | Two Step | MAME | LIME | Two Step | MAME |
| *Auto MPG* | 38.91 | **37.02** | *37.2* | 593.86 | *493.33* | **491.06** | 0.87 | **0.89** | **0.89** |
| *Retention* | 0.29 | **0.14** | **0.12** | 0.36 | *0.18* | **0.16** | *0.84* | **0.88** | 0.84 |
| *HELOC* | **0.05** | *0.06* | *0.06* | 0.14 | *0.11* | **0.10** | 0.94 | *0.95* | **0.96** |
| *Waveform* | 0.63 | *0.62* | **0.60** | **0.19** | 0.20 | **0.19** | 0.60 | **0.65** | *0.61* |
| *ATIS* | 0.19 | **0.17** | *0.18* | 0.37 | *0.14* | **0.12** | 0.90 | *0.90* | **0.91** |

(a)                                                                                           (b)

Table 1: Performance of the compared approaches. The first and second best numbers are respectively in **bold** and *italics*. The rows in (a) and (b) correspond to the same datasets. (a): Average infidelity measure [27] over all clusters/levels computed for the three compared methods with the black box models. Overall, MAME reduces explanation infidelity with respect to SP-LIME and Two Step by $11\%$ and $2\%$ for RF, and $34\%$ and $8\%$ for MLP. (b) Pearson correlation coefficient between black-box model feature importances (Random Forest) and feature importances of the explanation methods.

own train-test partition) and report mean performances. The black-box models trained are Random Forest (RF) Regressor/Classifier, and Multi-Layer Perceptron (MLP) Regressor/Classifier. Only the Auto MPG dataset used a regression black-box, while rest used classification black-box models. The RF models used between 100 and 500 trees, whereas the MLP models have either 3 or 4 hidden layers with 20 to 200 units per layer. For regression, the labels for the explanation models are directly the predictions, whereas for classification, these are the predicted probabilities for a specified class.

When running LIME and MAME, the neighborhood size $|\mathcal{N}_i|$ in (1) is set to 10, neighborhood weights $\psi(x_i, z)$ are set using a Gaussian kernel on $\|x_i - z\|_2^2$ with a automatically tuned bandwidth, and the $\alpha_i$ values in (1) are set to provide explanations with 5 non-zero values when $\beta = 0$. Tuning $\alpha_i$ for each instance happens only once in the leaf nodes and can be efficiently obtained using the LASSO homotopy method [31].

For Two Step and MAME, the labels $f(x_i)$ are sorted and the $w_{ij}$ (see Section 3) are set to 1 whenever $f(x_i)$ and $f(x_j)$ are right next to each other in this sorted list, else $w_{ij} = 0$. This path graph [32] prior enforces the simple observation that explanations for similar black-box model predictions must be similar, and provided good results. This prior knowledge graph biases the tree building process, particularly in the lower levels, since merges in those levels (Algorithm 1) follow the graph closely. So, it must be carefully created by the user, based on domain knowledge or experimentation. Another option, which we do not explore here, is to create the graph based on the instances $(x_i)$. This will perform well if there is a similarity measure under which similar instances would result in similar explanations. For both Two Step and MAME, we used post-processed explanations (see Section 3). More details on the datasets, the black-box models, the classes chosen for explanation, and other hyper-parameters are in the supplement.

The explanation infidelities are provided in Table 1 (a). The infidelities for Auto MPG dataset are higher since the regression outcomes can be much larger than 1 as opposed to other datasets where the classification probabilities are bound between 0 and 1. MAME is the first best in 7 out of 10 scenarios presented (5 datasets $\times$ 2 black-box models), and the second best in the rest 3. We also compute the relative reductions in infidelity across all datasets for the two black-box models. Overall, MAME reduces explanation infidelity with respect to SP-LIME and Two Step by $11\%$ and $2\%$ for RF, and $34\%$ and $8\%$ for MLP. . The feature importance correlation for RF black-boxes are provided in Table 1 (b). Here MAME performs competitively being the best in 3 and second best in the rest 2 datasets. We do not consider MLP for this measure since feature importances are not output by them. Overall, MAME performs better with these two measures because it creates more accurate approximations of black-box models at automatically-determined groupings of data.

For the three smaller datasets (Auto MPG, Retention, HELOC), all three methods complete running in about 10 minutes or less, in a single core. For Waveform, MAME and Two Step take $\sim 1.5$ hours to run whereas SP-LIME takes $\sim 5$ hours. For our largest dataset, ATIS, MAME and Two Step take $\sim 2$ hours and $\sim 3$ hours respectively to complete whereas SP-LIME takes $\sim 48$ hours. MAME and Two Step are thus naturally scalable for large, high-dimensional data, because of the underlying AR scheme. Additional details on computing infrastructure and computational complexity are given in the

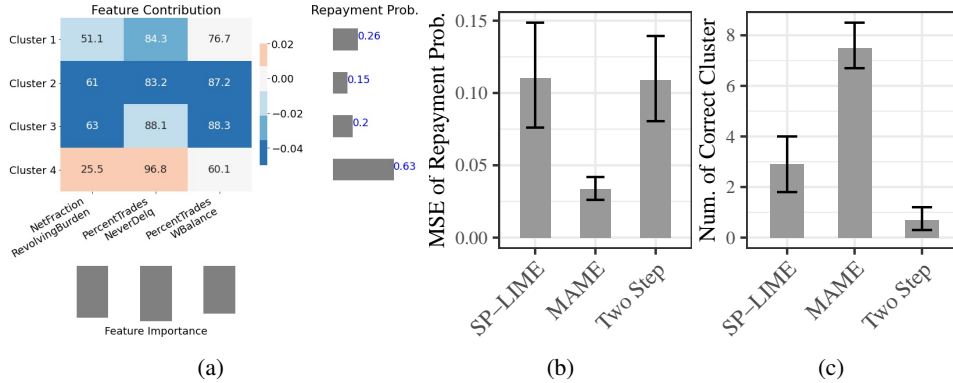

Figure 2: (a) The high-level visual explanation shown in the user study for the MAME condition. (b) Result of study: The left graph shows the mean squared error between the classifier's predicted repayment probability and the participants' guess of the classifier's prediction (lower is better). (c) Result of study: The average number of trials (out of 10) in which the participant correctly chose the most similar cluster (higher is better). The error bars show 95% confidence intervals.

supplement. Note that our implement was efficient, but we did not incorporate explicit parallelization, which can be used to further speed up the AR scheme.

## 4.3 User Study with Credit Dataset

To further evaluate MAME's ability to accurately capture high-level concepts for a given domain, we conducted another user study with 30 data scientists from three different technology companies. In this study, we asked the participants to play the role of a loan officer who needs to decide whether to approve or reject a person's loan request based on that person's financial features. We used the HELOC dataset [6] that contains 23 continuous variables about a person's financial record. One predictor, external credit score, was removed since it was a summary of other variables and accounted for much of the variability in the outcome variable. The binary outcome variable indicated either loan default or full repayment. We used 75% of the dataset to train a random forest model with 100 trees, and the remaining data for our user study. We then used MAME, SP-LIME, and Two Step to produce three different sets of explanations to characterize the training dataset. Figure 2 (a) shows the explanations generated by MAME (see Supplemental Material for a description of the visualization and the explanations generated by other methods). All methods were constrained to only produce four high-level explanation clusters so that they can be compared fairly and each cluster can represent a large portion of data.

The 30 participants were evenly divided into three groups, each exposed to only one explanation method, though they were not told what the method was. The participant's task is to review the explanations for the four clusters, assign the given 10 loan applications to the clusters, and guess the classifier's predicted loan repayment probability for these applications. In other words, we want the participants to simulate how the classifier make predictions, since the degree to which an algorithm can be simulated is an important metric to judge the efficacy of the explanations [33]. We expect the participants to do well in these tasks when the clusters produced by an explanation method are i) selective and ii) homogeneous (instances in each cluster have similar important feature values) and when the iii) explanations are faithful.

Figures 2 (b) and (c) shows the results of the user study. The left graph shows the mean squared error (MSE) between the classifier's probability prediction and the participant's guess of that prediction. A Tukey post hoc test shows that MAME ($\mu = 0.033$) significantly outperformed SP-LIME ($\mu = 0.11$, $p = .002$) and Two Step ($\mu = 0.109$, $p = .002$). The right graph shows the number of trials (out of 10) in which the participant chose the correct most-similar cluster. As can be seen, MAME ($\mu = 7.5$) again significantly outperformed SP-LIME ($\mu = 2.9$, $p < .0001$) and Two Step ($\mu = 0.7$, $p < .0001$). Both sets of results suggest that MAME produced more accurate and homogeneous high-level clusters than Two Step and SP-LIME methods. In addition, the very low squared difference suggests that non-experts can use MAME's high-level explanations to quickly understand how the model works and make reasonably accurate predictions.

### 4.4 Expert Study with Oil & Gas Industry Dataset

We perform a case study with a real-world industrial pump failure dataset (classification dataset) from the Oil & Gas industry. The pump failure dataset consists of sensor readings acquired from 2500 oil wells over a period of four years that contains pumps to push out oil. These sensor readings consist of measurements such as speed, torque, casing pressure (CPRESS), production (ROCURRRATE) and efficiency (ROCEFF) for the well along with the type of failure category diagnosed by the engineer. In this dataset, there are three major failure modes: Worn (pump is worn out because of aging), Reservoir (pump has a physical defect) and Pump Fit (there is sand in the pump). Semantically, there can be seven different types of pumps in a well which can be manufactured separately by fourteen different vendors. Our black-box classifier predicts the probability of Reservoir failure, which is a difficult failure class to identify. The black-box classifier used was a 7-layer multilayer perceptron (MLP) with parameter settings recommended by scikit-learn MLPClassifier. The dataset had 5000 instances 75% of which were used to train the model, and the rest were used for testing.

We conducted two types of studies here. In the first one, we obtained explanations from MAME without any side information from the subject matter expert (SME). The prior knowledge graph was just a *path graph* which is obtained by sorting the black-box predictions $f(x_i)$ and setting $w_{ij}$ to 1 whenever $f(x_i)$ and $f(x_j)$ are right next to each other in the sorted list. This simple prior (path graph) enforces that weights for similar observations must be similar. Relevant explanations are shown in Figure 1. They observed that at level 380, the two clusters shown (out of 4 at that level) were particularly interesting from a semantics perspective, as both corresponded to the same pump type, but had different manufacturer groups, and hence exhibited different behavior.

In the second study, we leveraged the SME's knowledge on the existence of 4 similar pump manufacturer groups in the data (mentioned above in study 1) and *induced this prior knowledge into MAME and Two Step* via the prior knowledge graph. Subsequently, the SME picked the level which had 4 clusters both with MAME and Two Step. The SME observed that the four clusters from MAME were completely homogeneous individually and captured the prior knowledge completely, whereas the four clusters from Two Step were heterogeneous individually and did not the reflect the expected semantic grouping. More explicitly, the SME concluded that MAME was able to ingest prior knowledge effectively and identify one cluster of non-producing pumps which were used in a run-to-failure scheme at higher speed, ROCURRRATE and CPRESS than usual. Two Step does not explicitly control for fidelity which results in less semantically relevant clusters. This helped the SME build more trust in our explanations. Both expert studies completed execution in 10 minutes or less, in a single core. Further details are available in the supplement.

## 5 Conclusion

In this paper, we have provided a meta-explanation method approach that can take a local explainer such as LIME and produce a multilevel explanation tree by jointly learning the explanations with different degrees of cohesion, while at the same time being faithful to the black-box model. We have argued based on recent works as well as through expert and non-expert user studies that such explanations can bring additional insight not conveyed readily by just global or local explanations. We have also shown that when one desires few explanations to understand the model and the data as typically would be the case, our method creates higher fidelity explanations compared with other methods. We have also made our algorithm scalable by proposing principled approximations to optimize the objective. Future extensions could involve adaptations to other non-parametric contrastive/counterfactual local explanation methods.

## Broader Impact

With the proliferation of deep learning, explaining or understanding the reasons behind the models decisions has become extremely important in many critical applications [30]. Many explainability methods have been proposed in literature [7, 12, 11], however, they either provide instance specific local explanations or fit to the entire dataset and create global explanations. Our proposed method is able to create both such explanations, but in addition, it also creates explanations for subgroups in the data and all of this jointly. We thus are creating explanations for granularities (between local and global). This multilevel aspect has not been sufficiently researched before. In fact recently

[4] has stressed the importance of having such multilevel explanations for successfully meeting the requirements of Europe's General Data Protection Regulation (GDPR) [5]. They clearly state that simply having local or global explanations may not be sufficient for providing satisfactory explanations in many cases.

There are also potential risks with this approach. The first is that if the base local explainer is non-robust or inaccurate [34, 35] then the explanations generated by our tree also may have to be considered cautiously. However this is not specific to our method, and applies to several post-hoc explainability methods that try to explain a black-box model. The way to mitigate this is to ensure that the local explanation methods are adapted (such as by choosing appropriate neighborhoods in LIME) to provide robust and accurate explanations. Another risk could be that such detailed multilevel explanations may reveal too much about the internals of the model (similar scenario for gradient-based models is discussed in [36]) and hence may raise privacy concerns. Mitigation could happen by selectively revealing the levels / pruning the tree or having a budget of explanations for each user to balance the level of explanations vs. the exposure of the black-box model.

## Funding and Conflicts of Interest

All authors were employed by IBM Corporation. There were no other sources of funding.

## Acknowledgments

The authors thank the anonymous reviewers for this and past revisions of this paper for their detailed and thoughtful comments. Part of this work was conducted under the auspices of the IBM Watson for Natural Resources Innovation Program.

## Footnotes

[1]We generated the data using code in `https://github.com/IBM/AIX360/blob/master/aix360/data/ted_data/GenerateData.py`

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
