[Supplementary Material]

# Supplement for
# Model Agnostic Multilevel Explanations

Karthikeyan Natesan Ramamurthy, Bhanukiran Vinzamuri, Yunfeng Zhang, Amit Dhurandhar

IBM Research, Yorktown Heights, NY USA 10598
`knatesa@us.ibm.com`, `bhanu.vinzamuri@ibm.com`, `zhangyun@us.ibm.com`, `adhuran@us.ibm.com`

The supplementary material provides additional details on experiments and theory discussed in the main paper. The figure and equation numbers are continued from the main paper.

## 1   Additional Details on Public Datasets

**Auto MPG:**   This dataset is obtained from `https://archive.ics.uci.edu/ml/datasets/Auto+MPG`. The features correspond to the various attributes of a model of a car, and the outcome is miles per gallon. This is the only regression dataset used.

**Retention:**   This is a synthetic dataset that is generated using `https://github.com/IBM/AIX360/blob/master/aix360/data/ted_data/GenerateData.py`. The features correspond to job position, organization, performance, compensation, and tenure of an employee and the outcome is whether the employee will leave the organization (1) or not (0). We choose to explain class 1.

**HELOC:** This dataset is obtained from `https://community.fico.com/s/explainable-machine-learning-challenge?tabset-3158a=2`.   The HELOC dataset had about 10000 instances, and some instances were basically empty and were excluded. From the rest, we used only the first 1000 instances both in the user study (Sections 4.3 and 4.2) in order to speed up explanation generation. The dataset had two labels - 0 indicating default on loan, and 1 indicating full repayment. We choose to explain class 1. The features correspond to financial health of people who apply for a loan.

**Waveform:**   The dataset was obtained from `https://archive.ics.uci.edu/ml/datasets/Waveform+Database+Generator+(Version+1)`. The outcome variable is 3 classes of waves (0, 1, 2) and the features are noisy combinations of 2 out of 3 base waves. We choose to explain class 0.

**ATIS:**   This dataset contains short texts along with 26 intents (labels) associated to each text, in a air travel information system.  This dataset was obtained from `https://www.kaggle.com/siddhadev/ms-cntk-atis` and processed using the code provided in `https://www.kaggle.com/siddhadev/atis-dataset-from-ms-cntk`. We used the slot filling IOB labels as input features after binarization - any value greater than 0 will be coded as 1. We also removed the last feature (O). The dataset is used for intent classification 26 intents and we choose to explain the class *flight*. We use the train and test partitions provided in the dataset itself, with 4978 and 893 examples respectively.

## 2   (Hyper-)Parameter Settings in Experiments

**Evaluation on Public Datasets and the User Study:**   The neighborhood size $|\mathcal{N}_i|$ was set to 10 when running LIME to generate the leaf explanations. We found this size to be reasonable since

larger sizes will make the codes run slower. We also found the default kernel width setting in LIME $(0.75 * \sqrt{p})$ worked well, so we used that. The number of non-zero coefficients in an explanation (a.k.a. explanation complexity) was set at 5, since we found it to be a good number for users to digest multiple explanations. The public datasets were evaluated using $5-$fold cross validation, except for ATIS which came with its own train-test partition. The training partitions were used to train the black box model and the explanation methods. Results were generated using the test partition,s and averaged.

When running MAME and Two Step, we set the multiplicative step-size $t$ to $1.01$, the initial regularization level for $\beta$, $\epsilon = 1e - 10$. We used $10$ conjugate gradient iterations when solving for $\Theta$ in (4).

**Evaluation for the Expert Study:** The neighborhood size $|\mathcal{N}_i|$ was set to $15$ here. Side information (refer Section 6 Study 2) from expert identifying pumps with the same manufacturer type consisted of 2155 edges. The explanation complexity for MAME was set at 3 (Figure 3). Figure 1 does not encode any side information and the complexity was set at 4. There were a total of 426 levels in the tree in Figure 1. The remaining parameters were set as mentioned above for the public datasets.

## 3   Computing Infrastructure

We ran the codes in an Ubuntu machine with 64 GB RAM and 32 cores. For the ATIS dataset alone, we used a machine with  250 GB RAM to avoid memory issues. Both MAME and Two Step were implemented without explicit parallelization, and the parallel operations only happened implicitly in Linear Algebra libraries. The codes were written in Julia 1.3 and also utilized some Python 3.7 libraries (e.g., for RF and MLP model building).

## 4   Note on Two Step Method Implementation

Since the optimization for Two Step method discussed in Section 3 is a special case of that of the MAME method in (1), we re-purposed the code written for MAME method by re-defining the variables and setting parameters appropriately.

## 5   Additional Evaluation on Public Datasets

In addition to using the explanation infidelity measure, we also use a generalized fidelity measure for understanding the trustworthiness of the explanations generated. Similar to the infidelity measure, we associate novel test examples to explanations, and then compute the $R^2$ of the prediction from the linear explanation model with respect to the black-box model's prediction. The $R^2$ value is the generalized fidelity. We compute this for MAME and Two Step for each level in the tree and compare the levels that have the same number of groups/clusters as the depth of the trees may vary. We also compute this measure for SP-LIME by varying the number of representative explanations from 1 to the total number of training examples. We remark that the $1-$nearest neighbor association is meaningful since LIME explanations (which also form the basis for Two Step and MAME) are optimized to work well in the neighborhood of each training data point.

The average generalized fidelity measures for the datasets for RF and MLP black-boxes are given in Table 1. Averaging is done in a similar way as the Explanation Infidelity measure discussed in Section 4.2(a). MAME is either the best or the second best in all cases. On an average, across all datasets, MAME improves generalized fidelity with respect to SP-LIME and two-step by $27\%$ and $14\%$ for RF, and $34\%$ and $10\%$ for MLP. In addition, we can see from the table that in 6 out of 10 datasets, SP-LIME produces negative $R^2$ values for at least one set of representative explanations, and this makes it an unreliable explanation method.

## 6   Expert Study with Oil & Gas Industry Dataset

We provide a full description of the case study described in Section 4.4 from the main paper.

| Dataset | Random Forest | | | MLP | | |
|---|---|---|---|---|---|---|
| | SP-LIME | Two Step | MAME | SP-LIME | Two Step | MAME |
| *Auto MPG* | *0.92* | *0.92* | **0.93** | 0.75 (0.68) | **0.84** | *0.83* |
| *Retention* | 0.22 (-0.03) | **0.49** | *0.48* | 0.25 (-0.03) | **0.41** | *0.40* |
| *HELOC* | **0.63** | 0.43 | *0.55* | 0.19 (0.16) | *0.21* | **0.25** |
| *Waveform* | **0.65** | 0.46 | *0.61* | *0.36* | 0.32 | **0.41** |
| *ATIS* | 0.50 (0.35) | *0.63* | **0.69** | 0.43 (0.25) | *0.63* | **0.67** |

Table 1: Average generalized fidelity measure over all clusters/levels computed for the three compared methods with the black box models. The first and second best numbers are respectively in **bold** and *italics*. The numbers in parenthesis represent true average $R^2$ taking negative values into account, whereas the ones outside are averages with negative set to 0. Overall, MAME improves generalized fidelity with respect to SP-LIME and Two Step by $27\%$ and $14\%$ for RF, and $34\%$ and $10\%$ for MLP.

We perform a case study with a real-world industrial pump failure dataset (classification dataset) from the Oil & Gas industry. The pump failure dataset consists of sensor readings acquired from 2500 oil wells over a period of four years that contains pumps to push out oil. These sensor readings consist of measurements such as speed, torque, casing pressure (CPRESS), production (ROCURRRATE) and efficiency (ROCEFF) for the well along with the type of failure category diagnosed by the engineer. In this dataset, there are three major failure modes: Worn, Reservoir and Pump Fit. Worn implies the pump is worn out because of aging. Reservoir implies the pump has a physical defect. Pump Fit implies there is sand in the pump. From a semantics perspective, there can be seven different types of pumps in a well which can be manufactured separately by fourteen different vendors. We are primarily interested in modeling reservoir failures as they are the more difficult class of failures to identify. The black-box classifier used was a 7-layer multilayer perceptron (MLP) with parameter settings recommended by scikit-learn MLPClassifier. The dataset had 5000 instances 75% of which were used to train the model and the remaining 25% was test.

We conducted two types of studies here. One where we obtained explanations without any side information from the experts and the other where we were told certain groups of pumps that should exhibit similar behavior and hence should have similar explanations for their outputs.

**Study 1- Expert Evaluation:** In this study, we built the MAME tree on the training instances. We picked a level which had 4 clusters guided by expert input given the dataset had 4 prominent pump manufacturer groups. In Figure 1 from the main paper, in the introduction, we show the root and example leaves and the level with 4 clusters (level 380), where two of these clusters are shown that the expert felt were semantically meaningful. The expert said that the two clusters had semantic meaning in that, although both clusters predominantly contained the same type of Progressive Cavity pump (main pump type of interest), they were produced by different manufacturers and hence, had somewhat different behaviors. Two of the manufacturers were known to have better producing pumps in general which corresponds to the explanation on the right compared to that on the left which had pumps from all other mediocre producing manufacturers, which was consistent with on-field observations. Hence, the result uncovered by MAME without any additional semantic information gave the expert more trust in the model that was built.

**Study 2- Expert Evaluation with Side Information:** In this study, the expert provided us a grouping of pumps that should have similar explanations. We incorporated this knowledge ($W$) (refer Eqn. (1) from main paper) into both MAME and Two Step (which also has the fusion term in its formulation). Based on expert input, we picked the level in the tree generated both by MAME and Two Step which had four clusters (as done in Study 1). The expert observed that none of the four clusters obtained from Two Step were homogeneously aligned with the four pump manufacturer groups (prior knowledge incorporated). However, two of the bigger homogeneous clusters in the MAME tree which are shown in Figure 3 were of interest to the expert. The expert provided the following insights for both these clusters: all *non-producing* (Well-down category) pumps in the homogeneous cluster in Figure 3g are more likely to be used in a run-to-failure scheme where they are used rigorously (i.e. at higher speed, ROCURRRATE and CPRESS) than they would be run regularly which explains why these factors impact reservoir type failures here.

All Producing pumps in the homogeneous cluster in Figure 3e on the other hand keep producing oil adequately for longer periods while operating at optimal efficiency (slightly lower than max efficiency). Also operating these producing pumps with lower torque (caused by the helical rotor

(a) Two Step Cluster 1

(b) Two Step Cluster 2

(c) Two Step Cluster 3

(d) Two Step Cluster 4

(e) MAME Cluster 1 (Producing pumps)

(f) MAME Cluster 2

(g) MAME Cluster 3 (Non-Producing pumps)

(h) MAME Cluster 4

Figure 3: Expert study on Oil & Gas Industry dataset depicting exemplar explanations for levels with four clusters using both MAME and Two Step with Side Information.

in the pump) can elongate their operational lifetime and reduce likelihood for imminent failure. These insights helped the expert to build more trust in MAME. Both these expert studies were also conducted on the same computing infrastructure described above, and they completed execution in 10 minutes or less in a single core.

## 7 Approximation Quality and Timing Comparisons between the Exact and AR-based methods

In order to demonstrate the approximation quality between Exact and AR-based solutions, we plot the approximate quality and timing between the two methods. Note that the exact method runs the ADMM iterations in (4)-(8) several times for each $\beta$ value until convergence, whereas the AR-based method runs the iterations only once for each $\beta$ value.

We use the *Auto MPG* dataset (both train and test partition together) to obtain a RF regressor black box model and train MAME trees. We set $\epsilon = 1e - 10$ and choose $t$ from

$\{1.01, 1.05, 1.1, 1.2, 1.3, 1.4, 1.5\}$. We therefore run the exact and AR-based methods 7 times each, one for each value of $t$. Both the exact and AR-based methods are warm-started using the solution for previous value of $\beta$. The approximation quality is measure by a normalized version of the measure given in Theorem 3.1. The normalization factor that divides the measure is $pn\mu$ where $\mu = \max_{i,j:(i,j)\in\mathcal{E}} \|\theta_i^{(0)} - \theta_j^{(0)}\|_2$. Note that the superscript $(0)$ indicates that the explanations belong to the leaf nodes.

From Figure 4, we see that the exact and approximate solutions get closer as $t \to 1$ as predicted by the theory. We also see from Figure 5 that the AR-based solution is around 10 times faster to compute than the exact solution for all $t$ values. Both these results demonstrate the utility of the AR-based approximate method.

Figure 4: Normalized distance between exact and AR-based solutions.

Figure 5: Runtime comparison between exact and AR-based solutions.

# 8 Proof Sketch of Lemma 3.1

*Proof Sketch.* If $O_p$ denotes the objective in equation 1 being optimized at level $p$, then $O_p = O_{p-1} + \Delta_p \left( \sum_{i<j} w_{ij} \|\theta_i - \theta_j\|_2 \right)$, where $\Delta_p = \beta_p - \beta_{p-1}$. We know that $\Delta_p > 0$ by design and so we have an added penalty.

If at the optimal of level $p$, $\|\theta_i^{(p)} - \theta_j^{(p)}\|_2 > \|\theta_i^{(p-1)} - \theta_j^{(p-1)}\|_2$ for some $x_i$ and $x_j$, then that would imply that the other two terms in the objective reduce enough to compensate for the added penalty. However, this would imply that $\theta_i^{(p-1)}$ and $\theta_j^{(p-1)}$ were not the optimal solution at level $p-1$ as the current solution would be better given the lesser emphasis on the last term (i.e. lesser $\beta$) at that level. This contradicts our assumption. $\qquad\square$

# 9 Linear Convergence of ADMM for MAME

Strong convexity of the objective function ensures linear convergence of the ADMM method in practice. However, in the absence of strong convexity certain additional criteria need to be satisfied to have linear convergence. Hong and Luo in their paper [1] established the global linear convergence of ADMM for minimizing the sum of any number of convex separable functions expressed in the form below.

$$\text{minimze} \quad f(x) = f_1(x_1) + f_2(x_2) + \ldots + f_K(x_K) \tag{10}$$
$$\text{subject to} \quad Ex = E_1 x_1 + E_2 x_2 + \ldots + E_K x_K = q$$
$$x_k \in X_k, k = 1, 2, \ldots, K$$

The major assumptions imposed by the authors in their paper to prove the linear convergence for the function $f(x)$ are the following

1. Each $f_k$ can further be decomposed as $f_k(x_k) = g_k(A_k x_k) + h_k(x_k)$ where $g_k$ and $h_k$ are both convex and continuous over their domains.

2. Each $g_k$ is strictly convex and continuously differentiable with a uniform Lipschitz continuous gradient.

$$\|\nabla A_k^T g_k(Ax_k) - A_k^T \nabla g_k(Ax_k')\| \le L\|x_k - x_k'\|$$

3. Each $h_k$ satisfies either one of the following conditions
   - The epigraph of $h_k(x_k)$ is a polyhedral set.
   - $h_k(x_k) = \lambda_k \|x_k\|_1 + \sum_J w_J \|x_{k,J}\|_2$ where $x_k = (\ldots, x_{k,J}, \ldots)$ is a partition of $x_k$ with $J$ being the partition index.
   - Each $h_k(x_k)$ is the sum of the functions described in the previous two terms.

4. For any fixed and finite y and $\xi$, $\sum_k h_k(x_k)$ is finite for all $x \in \{x : L(x; y) \le \xi\} \cap X$.

5. Each submatrix $E_k$ has full column rank.

6. The feasible sets $X_k, k = 1, \ldots, K$ are compact polydedral sets.

In Equation 10, each $f_k$ is a convex function subject to linear equality constraints. Our original MAME problem in Equation 2 can be written in the form below as in Equation 11, which satisfies all the necessary criteria specified above from 1-6. For example, it is known that the epigraph of the $\ell_1$ norm is a polydedral set. The I identity matrix and the difference matrix $D$ satisfy the full column rank condition also.

Hong and Luo [1] additionally mentions that each $f_k$ may only consist of the convex non-smooth function $h_k$ and the strongly convex part $g_k$ can be absent. This helps us ensure that the two sparsity inducing norm terms in our formulation on $U$ and $V$ do not violate any of the conditions stated. To understand the proof of convergence, we recommend the readers to go through the proof provided in Hong and Luo [1]. This completes the proof for linear convergence of ADMM for MAME.

$$\underset{\Theta, U, V}{\arg \min} \sum_{i=1}^{n} \sum_{z \in \mathcal{N}_i} \psi(x_i, z) \left( f(z) - g(z)^T \theta_i \right)^2 \tag{11}$$

$$+ \alpha_i \|U_{.i}\|_1 + \beta \left( \sum_{e_l \in \mathcal{E}} w_l \|V_{.l}\|_2 \right),$$

$$\text{such that} \quad \Theta(I + D) - IU - IV = 0,$$

# 10 Proof of Theorem 3.1 (from main paper)

In this section we prove Theorem 3.1 on the expectation of difference between exact and AR solutions. We begin with 3 technical lemmas based on the lemmas given in [2] which maybe of independent interest: Lemma 10.1 provides a convergence rate for the optimization step embedded within an iteration; Lemma 10.2 establishes a form of Lipschitz continuity for convex clustering regularization paths; Lemma 10.3 provides a global bound for the approximation error induced at any iteration.

**Lemma 10.1** (Q-Linear Error Decrease). *At each iteration $k$, the approximation error decreases by a factor $c < 1$ not depending on $t$ or $\epsilon$. That is,*

$$\|\Theta^{(k)} - \Theta^{(\beta)}\| < c\left[\|\Theta^{(k-1)} - \Theta^{(\beta)}\|\right]$$

*for some $c$ strictly less than 1 where $\beta$ is the computed regularization parameter on the AR path $(\beta = \gamma^{(k)})$.*

*Proof.* In the notation of [1], the constraint matrix for MAME problem from Equation 11 is given by $E = (I + D \quad -I \quad -I)$, for appropriately sized identity matrices, which is clearly row-independent (one of the assumptions mentioned in Section 9), yielding linear convergence of the primal and dual variables at a rate $c_\lambda < 1$ which may depend on $\lambda$. This follows from the proof sketch for the linear convergence of ADMM for MAME which has been provided in Section 9 where we show how our formulation satisfies all the assumptions stated in [1]. Taking $c = \sup_{\lambda \le \lambda_{\max}} c_\lambda$, we observe that the MAME iterates are uniformly Q-linearly convergent at a rate $c$. □

**Lemma 10.2** (Lipschitz Continuity of Solution Path). $\Theta^{(\lambda)}$ *is L-Lipschitz with respect to $\lambda$. That is,*

$$\|\Theta^{(\lambda_1)} - \Theta^{(\lambda_2)}\| \le L * |\lambda_1 - \lambda_2|$$

*for some $L > 0$.*

*Proof.* We first show that $\Theta^{(\lambda)}$ is Lipschitz. The vectorized version of MAME problem can be written as

$$\theta^{(\lambda)} = \arg\min_{\theta \in \mathbb{R}^{n \times p}} \frac{1}{2}\|\theta - u\|_2^2 + \lambda f_p(\theta) + \lambda f_q(\tilde{D}\theta)$$

where $u = \text{vec}(\psi^{1/2} f(z))$, $\theta = \text{vec}(\psi^{1/2} g(z)^T \Theta)$, $f_q$ and $f_p$ are convex functions, and $\tilde{D} = I \otimes D$ is a fixed matrix. The KKT conditions give

$$0 \in \theta_\lambda - u + \lambda \partial f_p(\theta_\lambda) + \lambda \tilde{D}^T \partial f_q(\tilde{D}\theta_\lambda)$$

where $\partial f_p(\cdot)$, $\partial f_q(\cdot)$ are the subdifferential of $f_p$ and $f_q$. Since both $f_p$ and $f_q$ are convex, it is differentiable almost everywhere [Theorem 25.5] [3], so the following holds for almost all $\theta_\lambda$:

$$0 = \theta_\lambda - u + \lambda f_p'(\theta_\lambda) + \lambda \tilde{D}^T f_q'(\tilde{D}\theta_\lambda)$$

Differentiating with respect to $\lambda$, we obtain

$$0 = \theta_\lambda - u + \lambda f_p'(\theta_\lambda) + \lambda \tilde{D}^T f_q'(\tilde{D}\theta_\lambda)$$

$$\frac{\partial}{\partial \lambda}[0] = \frac{\partial}{\partial \lambda}\left[\theta_\lambda - u + \alpha f_p'(\theta_\lambda) + \beta \tilde{D}^T f_q'(\tilde{D}\theta_\lambda)\right]$$

$$0 = \frac{\partial \theta_\lambda}{\partial \lambda} - 0 + \lambda \frac{\partial}{\partial \lambda}\left[f_p'(\theta_\lambda)\right] + f_p'(\theta_\lambda)$$

$$+ \lambda \frac{\partial}{\partial \lambda}\left[\tilde{D}^T f_q'(\tilde{D}\theta_\lambda)\right] + \tilde{D}^T f_q'(\tilde{D}\theta_\lambda)$$

$$0 = \frac{\partial \theta_\lambda}{\partial \lambda} + \lambda f_p''(\theta_\lambda)\frac{\partial \theta_\lambda}{\partial \lambda} + f_p'(\theta_\lambda)$$

$$+ \lambda \tilde{D}^T f_q''(\tilde{D}\theta_\lambda)\tilde{D}\frac{\partial \theta_\lambda}{\partial \lambda} + \tilde{D}^T f_q'(\tilde{D}\theta_\lambda)$$

$$\implies \frac{\partial \theta}{\partial \lambda} = -[I + \lambda f_p''(\theta)]^{-1} f_p'(\theta)$$

$$- [I + \lambda \tilde{D}^T f_q''(\tilde{D}\theta)\tilde{D}]^{-1} D^T f_q'(\tilde{D}\theta).$$

Note that $\theta_\lambda$ depends on $\lambda$ so the chain rule must be used here. From here, we note

$$\left\|\frac{\partial \theta_\lambda}{\partial \lambda}\right\|_\infty$$

$$\le \| - [I + 0]^{-1} f_p'(\theta_\lambda) - [I + 0]^{-1} \tilde{D}^T f_q'(\tilde{D}\theta_\lambda)\|_\infty$$
$$= \|f_p'(\theta_\lambda) + \tilde{D}^T f_q'(\tilde{D}\theta_\lambda)\|_\infty.$$

For the MAME problem, we recall that $f_p(\cdot)$ and $f_q(\cdot)$ are convex norms and hence have bounded gradients; hence $f_p'(\theta_\lambda)$ and $f_q'(\tilde{D}\theta_\lambda)$ are bounded so the gradient of the regularization path is bounded and exists almost everywhere. This implies that the regularization path is *piecewise* Lipschitz. Since the solution path is constant for $\lambda \geq \lambda_{\max}$ and is continuous, the solution path is globally Lipschitz with a Lipschitz modulus equal to the maximum of the piecewise Lipschitz moduli. $\qquad \square$

**Lemma 10.3** (Global Error Bound). *The following error bound holds for all $k$:*

$$\|\Theta^{(k)} - \Theta^{(\beta)}\| \leq c^k L\epsilon + L(t-1)\epsilon t^k \sum_{i=1}^{k-1} \left(\frac{c}{t}\right)^i$$

*Proof.* Our proof proceeds by induction on $k$. First note that, at initialization:

$$\|\Theta^{(0)} - \Theta^{(\epsilon)}\| \leq L\epsilon$$

by Lemma 10.2.

Next, at $k = 1$, we note that

$$\|\Theta^{(1)} - \Theta^{(t\epsilon)}\| \leq c\|\Theta^{(0)} - \Theta^{(t\epsilon)}\|$$

by Lemma 10.1. We now use the triangle inequality to split the right hand side:

$$\|\Theta^{(0)} - \Theta^{(t\epsilon)}\| \leq \underbrace{\|\Theta^{(0)} - \Theta^{(\epsilon)}\|}_{\text{RHS-1}} + \underbrace{\|\Theta^{(\epsilon)} - \Theta^{(t\epsilon)}\|}_{\text{RHS-2}}$$

From above, we have RHS-1 $\leq L\epsilon$. Using Lemma 10.2, RHS-2 can be bounded by

$$\|\Theta^{(\epsilon)} - \Theta^{(t\epsilon)}\| \leq L\,|t\epsilon - \epsilon| = L(t-1)\epsilon.$$

Putting these together, we get

$$\|\Theta^{(1)} - \Theta^{(t\epsilon)}\| \leq c\,[\text{RHS-1} + \text{RHS-2}] \leq c\,[L\epsilon + L(t-1)\epsilon] = cLt\epsilon$$

Repeating this argument for $k = 2$, we see

$$
\begin{aligned}
\|\Theta^{(2)} - \Theta^{(t^2\epsilon)}\| &\leq c\|\Theta^{(1)} - \Theta^{(t^2\epsilon)}\| \\
&\leq c\left[\|\Theta^{(1)} - \Theta^{(t\epsilon)}\| + \|\Theta^{(t\epsilon)} - \Theta^{(t^2\epsilon)}\|\right] \\
&\leq c\left[cLt\epsilon + L\,|t^2\epsilon - t\epsilon|\right] \\
&= c^2 Lt\epsilon + cL(t-1)\epsilon * t \\
&= c^2 Lt\epsilon + L\epsilon(t-1)t^2 * \left(\frac{c}{t}\right) \\
&= c^2 Lt\epsilon + L\epsilon(t-1)t^2 * \sum_{i=1}^{k-1}\left(\frac{c}{t}\right)^i
\end{aligned}
$$

We use this as a base case for our inductive proof and prove the general case:

$$
\begin{aligned}
\|\Theta^{(k)} - \Theta^{(t^k\epsilon)}\| &\leq c\|\Theta^{(k-1)} - \Theta^{(t^k\epsilon)}\| \\
&\leq c\left[\|\Theta^{(k-1)} - \Theta^{(t^{k-1}\epsilon)}\| + \|\Theta^{(t^{k-1}\epsilon)} - \Theta^{(t^k\epsilon)}\|\right] \\
&\leq c\left[c^{k-1}Lt\epsilon + L\epsilon(t-1)t^{k-1}\sum_{i=1}^{k-2}\left(\frac{c}{t}\right)^i + L\,|t^k\epsilon - t^{k-1}\epsilon|\right] \\
&= c^k Lt\epsilon + cL\epsilon(t-1)t^{k-1}\sum_{i=1}^{k-2}\left(\frac{c}{t}\right)^i + cL\epsilon(t^k - t^{k-1}) \\
&= c^k Lt\epsilon + L\epsilon(t-1)t^k\left[\frac{c}{t}\sum_{i=1}^{k-2}\left(\frac{c}{t}\right)^i + \frac{c}{t}\right]
\end{aligned}
$$

$$= c^k L t \epsilon + L\epsilon(t-1)t^k \left[ \sum_{i=2}^{k-1} \left(\frac{c}{t}\right)^i + \frac{c}{t} \right]$$

$$= c^k L t \epsilon + L\epsilon(t-1)t^k \sum_{i=1}^{k-1} \left(\frac{c}{t}\right)^i$$

With these results, we are now ready to prove Theorem 3.1 $\qquad \square$

*Proof.* We begin by fixing temporarily $\beta$ and bounding

$$\inf_k \left\| \Theta^{(k)} - \Theta^{(\beta)} \right\|$$

The infimum over all $k$ is less than the distance at any particular $k$, so it suffices to choose a value of $k$ which gives convergence to 0. Let $\tilde{k}$ be the value of $k$ which gives the closest value of $\gamma^{(k)}$ to $\beta$ along the AR path; and let $\tilde{\beta} = \gamma^{(\tilde{k})} = \epsilon t^{\tilde{k}}$. That is,

$$\tilde{k} = \arg\min_k |\gamma^{(k)} - \beta| \quad \text{and} \quad \tilde{\beta} = \gamma^{(\tilde{k})}$$

Then

$$\inf_k \left\| \Theta^{(k)} - \Theta^{(\beta)} \right\| \leq \| \Theta^{(\tilde{k})} - \Theta^{(\beta)} \| \leq \underbrace{\| \Theta^{(\tilde{k})} - \Theta^{(\tilde{\beta})} \|}_{\text{RHS-1}} + \underbrace{\| \Theta^{(\tilde{\beta})} - \Theta^{(\beta)} \|}_{\text{RHS-2}}$$

Using Lemma 10.2, we can bound RHS-2 as

$$\text{RHS-2} \leq L|\tilde{\beta} - \beta| \leq L|\gamma^{(\tilde{k}+1)} - \gamma^{(\tilde{k}-1)}| = L * \epsilon t^{\tilde{k}-1} * [t^2 - 1]$$
$$\leq L * \beta_{\max} * [t^2 - 1]$$

Using Lemma 10.3, we can bound RHS-1 as

$$\text{RHS-1} \leq c^{\tilde{k}} L\epsilon + L(t-1) * \epsilon t^{\tilde{k}} \sum_{i=1}^{k-1} \left(\frac{c}{t}\right)^i$$
$$\leq c^{\tilde{k}} L\epsilon + L(t-1) * \epsilon t^{\tilde{k}} * C$$

where $C = \sum_{i=1}^{\infty} \left(\frac{c}{1+t}\right)^i$ is large but finite. Since $c < 1$ and $\tilde{\beta} = \epsilon t^{\tilde{k}} \leq \beta_{\max}$, we can replace the $k$-dependent quantities to get

$$\text{RHS-1} = \| \Theta^{(\tilde{k})} - \Theta^{(\tilde{\beta})} \| \leq L\epsilon + C * L(t-1) * \beta_{\max}$$

Putting these together, we have

$$\inf_k \| \Theta^{(k)} - \Theta^{(\beta)} \| \leq \text{RHS-1} + \text{RHS-2}$$

$$\leq L\epsilon + C * L(t-1) * \beta_{\max} + L * \beta_{\max} * [t^2 - 1]$$

Similarly by fixing $k$ we know that

$$\inf_\beta \| \Theta^{(k)} - \Theta^{(\beta)} \| \leq \| \Theta^{(k)} - \Theta^{(\tilde{\beta})} \|$$

where

$$\tilde{\beta} = \arg\min_\beta |\log_\gamma \beta - k| \quad \text{and} \quad \tilde{k} = \log_\gamma \tilde{\beta}$$

Using similar arguments as above based on Lemma 10.3 we can show that

$$\inf_\beta \| \Theta^{(k)} - \Theta^{(\beta)} \| \leq \| \Theta^{(k)} - \Theta^{(\tilde{\beta})} \| \leq L\epsilon + C * L(t-1) * \beta_{\max}$$

We can use these two bounds and plug them in here

$$\max \left\{ E_\beta \left( \inf_k \|\Theta^{(k)} - \Theta^{(\beta)}\| \right), E_k \left( \inf_\beta \|\Theta^{(k)} - \Theta^{(\beta)}\| \right) \right\}$$

$$\leq \max \left\{ E_\beta \left( L\epsilon + C * L(t-1) * \beta_{\max} + L * \beta_{\max} * [t^2 - 1] \right), \right.$$

$$\left. E_k \left( L\epsilon + C * L(t-1) * \beta_{\max} \| \right) \right\} \xrightarrow{(t,\epsilon)\to(1,0)} 0$$

One can observe that as $t, \epsilon \to$ to $(1, 0)$ both the expectation terms reduce to 0 individually. $\qquad \square$

## 11  User Study Material

Figure 6 is a screenshot of the user study in the LIME condition. The visual explanations show the feature contribution, likelihood of repayment, and feature importance calculated using the SP-LIME method. The top left graph shows that the classifier groups loan applicants into four clusters (y-axis) based on several key features (x-axis). The color of the squares indicates the average contribution of a feature to the classifier's prediction for a cluster, with orange colors indicating increasing probability of full repayment, whereas blue indicating decreasing probability. The numbers in the colored squares show the average feature value for that cluster. The top right graph shows the average of the classifier's predicted probability of loan repayment of each cluster. The bottom graph shows the overall importance of the features used in the top left graph. The longer the bar, the more important it is to the classifier. The same set of visual explanations was used throughout each condition as it does not change from trial to trial.

Below the explanation is a table showing the 22 financial records for a loan applicant. It is divided into two segments. The top segment shows the features used in the visual explanation, which are also the most important features deemed by the explanation algorithm. The bottom segment shows the rest of the features. At the bottom of the page are two questions that the participant had to answer for every trial. The participant could access a written instruction any time during the experiment by clicking the "SEE INSTRUCTIONS" button at the top of the web page.

## 12  Computational Complexity Analysis

We provide the computational complexity for running one set of iterations given in (4)-(8). This is the same as obtaining solutions for one $\beta$ value if we use the AR-based method. Let us consider the five steps individually.

For (4), we use conjugate gradients to obtain the solution. If we assume the number of edges $|\mathcal{E}| = O(n)$, and the number of CG iterations to be $s$, the dominant complexity of this step is $O(p^2 n s) + O(p n^2 s)$. For (5), the update involves a soft-thresholding step which incurs a complexity of $O(pn)$. The update step (6), similarly incurs a complexity of $O(pn)$, assuming $|\mathcal{E}| = O(n)$. Updates for $Z_1$ and $Z_2$ in (7) and (8) respectively involve complexities of $O(pn)$ each. If we assume $s$ to be very small (we use 10 in our experiments), the dominant complexity for one ADMM iteration is hence $O(pn(p + n))$.

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

SEE INSTRUCTIONS

**For a person with the following financial record:**

\* -7 indicates no inquiries or no delinquencies and -8 indicates no usable/valid trades or inquiries.

\* Hover on the feature names to reveal feature descriptions.

**Important Features**

| | | |
|---|---|---|
| MSinceMostRecentTradeOpen : 9 | MaxDelq2PublicRecLast12M : 7 | NetFractionInstallBurden : -8 |
| NetFractionRevolvingBurden : 0 | PercentTradesNeverDelq : 100 | |

**Other Features**

| | | |
|---|---|---|
| AverageMInFile : 68 | MSinceMostRecentDelq : -7 | MSinceMostRecentInqexcl7days : -7 |
| MSinceOldestTradeOpen : 202 | MaxDelqEver : 8 | NumBank2NatlTradesWHighUtilization : 0 |
| NumInqLast6M : 0 | NumInqLast6Mexcl7days : 0 | NumInstallTradesWBalance : 2 |
| NumRevolvingTradesWBalance : 0 | NumSatisfactoryTrades : 21 | NumTotalTrades : 21 |
| NumTrades60Ever2DerogPubRec : 0 | NumTrades90Ever2DerogPubRec : 0 | NumTradesOpeninLast12M : 1 |
| PercentInstallTrades : 29 | PercentTradesWBalance : 25 | |

**Which cluster is the person most similar to?**

◯ Cluster 1     ⦿ Cluster 2     ◯ Cluster 3     ◯ Cluster 4

**What do you think is the classifier's predicted likelihood of repayment?**

65

0                                                          100

NEXT

Figure 6: A screenshot of the user study in the LIME condition.