[Reviews · NeurIPS 2020]

Review 1

Summary and Contributions: The paper proposes a new algorithm to create explanations that explain at multiple levels, such as individual instances, groups of instance, and overall model behavior. The algorithm is in fact a meta algorithm--- it can take existing post hoc explainers such as LIME to create these multi-level explanations. Authors achieve this by deriving an objective function and an optimization process for this objective. The objective function additionally allows accommodating prior knowledge of human experts for deciding the levels. Through qualitative experiments on multiple datasets and multiple user studies with 30 data scientist and domain experts, authors show that their approach produces higher fidelity explanations, allows data scientist to better simulate the model's predictions, and generates explanations that align with domain experts expectations.

Strengths: - Clear formulation of the objective function and optimization algorithm for creating multi-level explanations - Qualitative experiments on multiple datasets that show that algorithm generates higher fidelity solutions than prior work - Multiple user studies showcasing improved performance of data scientist at simulating model predictions and improved performance at gaining trust of domain experts.

Weaknesses: Please see my complete notes in additional feedback for details. - Does not discuss the limitations of the metric of infidelity considered in this work - Lacks discussion of how their system resulted in better user performance for both user studies with data scientist and domain experts.

Correctness: Yes, I believe it is correct. But authors do need to make the text, especially the experiments more clear-- please see my complete notes in the additional feedback section.

Clarity: The details of the user studies have a massive scope for improvement.

Relation to Prior Work: Mostly yes. But I pointed to two paper in my notes. Please see them.

Reproducibility: Yes

Additional Feedback: Fig 1. What does level mean here? Depth of the tree? How did experts figure out the cluster labels? Was the manufacturer a feature in the dataset? Or by manually inspecting the examples in the cluster? “Moreover, they can also provide exemplar based explanations looking at the groupings at specific levels (viz. different pumps by the same manufacturer)” This is not shown in Fig 1, but rather an additional affordance of your proposal? “The drawback here is that the explanations at levels other than local may not have high fidelity to the black-box” Isn’t this expected with any such approach including yours, i.e., wouldn’t any local-explanation lose fidelity as it explains a larger number of examples? For example, because the local model will increasingly be a poor approximation of models’ behavior, especially as it tries to capture more and more of mode’s global behavior (e.g., on groups of examples). Equation 1. Possibly bracket the summation on z? Also, how are weights generated in your experiments? As you correctly admit, users can specify this. But what would the algorithm do in the absence of experts prior knowledge? Experiments/Baselines. Related work mentioned work by Pedreschi et al. (2019) as similar, in that it generates a multi-level tree explanation like this work, then why was wasn’t it included in the baselines? I am sure the authors had a good reason, I just want them to clarify this in the text in Experiments or Related Work. Description of the Two Step baseline: What is p? What is F? What do tildas denote on all parameters? Ideally, this objective should be described using the same notation as equation 1 so that its to compare the differences. Why is SP-LIME a relevant baseline? Perhaps because it attempts to provide global explanations by outputting a set of (but not a tree) of local explanations. Authors may find this paper useful and relevant to cite: Lakkaraju, H., Kamar, E., Caruana, R., & Leskovec, J. (2019, January). Faithful and customizable explanations of black box models. In Proceedings of the 2019 AAAI/ACM Conference on AI, Ethics, and Society (pp. 131-138). Important missing detail (sorry if I missed!): How did you control for the amount each method explained? If this is not controlled, a method that explains more can get lower infidelity? For example, in the user study the number of cluster (=4) were controlled for. How was this done in the experiments presented in Table 1? What is the intuition for the correctness of specific definitions of infidelity in 4.2? In addition to the fact that it was proposed by previous work. A sentence or two would help. Since improved fidelity is a major contribution of this work, are there any limitations of the definition of fidelity considered here? If predictions of a local explainer and underlying model match, can we truly say their explanations will also match? Authors may find this paper useful to cite and describe the scope of their definition of fidelity: Jacovi, A., & Goldberg, Y. (2020). Towards Faithfully Interpretable NLP Systems: How should we define and evaluate faithfulness?. arXiv preprint arXiv:2004.03685. Table 1. Why is the range of values of Auto MPG so much higher? Because the range of the outcome variable was beyond 1? How stable are these results? Would recommend adding details about how many runs the results were averaged over and confidence intervals. The user study setup and goal is unclear to me. Were the clusters common across three systems? Why is asking users to which cluster an example belongs to, the correct metric? What did the explanations for other systems look like incomparsion? I realize that authors may have run out of space for adding these details. But the way it is presented right now, left me with many questions. (There is some redundant text between caption of Fig 2 and para at line 253, which authors may merge to make little more space) Figure 2a. What are the takeaways from this explanation in a few sentences? What is the subgraph underneath the grid? Importantly, why do authors intuitively think users were better able to simulate the model using their explanations? “The prior knowledge graph was just a path graph based on sorted prediction probabilities.” Unclear… “They observed that at level 380, the two clusters shown (out of 4 at that level) were particularly interesting from a semantics perspective, as both corresponded to the same pump type, but had different manufacturers, and hence exhibited different behavior.” Again, it is unclear what “level” means here and why there were 4 clusters at this “level.” Was this the only pattern they found that aligned with their prior experience? “The SME observed that the four clusters from MAME were completely homogeneous individually and captured the prior knowledge completely, whereas the four clusters from Two Step were heterogeneous individually and did not reflect the expected semantic grouping. This helped the expert build more trust in our explanations. Further details are available in the supplement.“ This seems like an important and exciting qualitative result. But I unfortunately had a hard time understanding from the main paper why this happened, e.g., what aspect of MAME enables this behavior that translated into end user studies. POST REBUTTAL -------------------- I thank the authors for answering so many of my questions!


Review 2

Summary and Contributions: Provide a way for feature attribution based explanations for single data-points as well as groups of closely related points. This is done by a cost function that encourages nearby points to have similar or same explanation vectors via a regularization term, invoking ideas from convex clustering.

Strengths: Multi-level explanatiions are relatively less explored in comparison to point explanations or global ones. Nice user study. Also a nice real life application, even though the dataset is not public.

Weaknesses: There are several math issues in the paper. for example, local model is a GAM (using g()), not never specified how g() is obtained, and in particular the optimization in Eq. 1 or 2 doesn't involve g. Looks like there is a hyper-parameter (alpha) for every data point, That is strange and seems highly inefficient. What if all the alphas were the same? Why is L1 penalty not used in the 3rd term of Eq. 1, it seems more natural, though then the optimization does not directly port over from prior work. Also the evaluation part is not compelling.

Correctness: See above. re: results: it is a mixed bag. For example, for Waveform it is really poor; that would be OK if there was insight/explanation, but I could not find it. Not clear why N_i is fixed and how to determine it.

Clarity: quite good, except for the math issues.

Relation to Prior Work: yes in general.MAPLE (ref 13) ws mentioned in passing, but as it is quite relevant and similar in intent, I would have liked a much closer comparison/discussion with this prior work.

Reproducibility: Yes

Additional Feedback: UPDATE: I have read and taken into account the rebuttal, as well as any ensuing discussions.


Review 3

Summary and Contributions: After reading rebuttals and other reviews, I still think the required prior graph might not be that useful in the application. And as R3 mentions, the potentially long computation of lambda_i per example could be time consuming. I still remain marginally above the threshold. ========= Original review ========== This paper presents a method that explains a black box model with multiple hierachical groupings such that each group has a linear weights vectors for each feature. They first explain each example by LIME (i.e. using a linear model to approximate the behavior of the black box in the perturbation neighborhood). In addition to l1 sparsity penalty for explanations, they also add l2 fusing penalty across examples for the LIME weights (the penalty graph could be humanly defined or based on the similarity of the model's output). And they slowly increase the l2 fusing penalty until there is only 1 grouping as global explanation.

Strengths: The method is novel to be applied in the explainability space and the optimization method is interesting and non-trivial. The experiments are comprehensive and do several human studies. The baselines are well designed.

Weaknesses: 1) Although this method is conceptually solid, the experimental results seem to be close to the two other simpler baselines especially "Two Step". From Table 1 I feel that Two Step actually do a very good job already and MAME seems to have not much difference. Especially there are some hyperparameters choices (e.g. the sparsity penalty that keeps only 5 features) that might affect the result. 2) I appreciate there are two human studies (4.3, 4.4). But I don't understand intuitively why users can guess model's output much better in MAME (Fig. 2b), even after seeing the visualization interface in the supplementary. Is it because the user can guess which cluster it belongs to more easily? But isn't this method supposed to tell the user which cluster this examples already? I guess I don't know why the user need to guess which cluster it belongs to? Also, in Sec. 4.4 (line 284), "the SME observed that ... MAME were completely homogenous individually and captured the prior knowledge completely ...". Is it possible to have a qualitative example of what this means and why this is important? (3) The previous two points make me wonder how useful this method would be in the real world (similar to two-step baseline and no intuitive understanding of why practioners would find it useful). I also think the biggest Achilles heel for this paper is the required existence of the prior knowledge graph, which is usually unavailable in the real world data. Also, sometimes the prior knowledge graph might not be valid in the data. If that's the case, will this method produce something wrong but because of its homogeneity that misleads the user to trust it more?

Correctness: Yes

Clarity: Yes. Some minor suggestions are maybe the authors can move algorithm 1 box closer to its description in page 4 or 5 (it's a bit painful to go back and forth). Also, how does the author learn g(z) in equation (2)? I feel that g(z) is probably just z for continuous feature and one-hot encoding value for categorical feature z?

Relation to Prior Work: Yes.

Reproducibility: Yes

Additional Feedback: Overall I enjoy reading the methodology and experiments, but I probably feel this method is not that useful in practice. Also, the lack of novelty seems to be slightly below the bar.


Review 4

Summary and Contributions: The paper presents a novel approach to generate a tree of explanations for a given classifier or regressor f, in the sense that local explanations for the output of f at each training example x form the bottom of the tree and higher levels are formed by explanations for clusters of data points until we arrive at the root with an explanation of the classifier for all datapoints. An explanation, in this framework, is a linear approximation of f, i.e. an explanation is a vector \theta such that f(x) is approximately equal to \theta * x. The bottom of the tree is formed by training an individual \theta for each data point which approximates f in the local neighborhood of f. Then, one increases regularization strength to make \theta vectors more similar and merges examples into clusters with similar \theta vectors until all points are in the same cluster (which is then the root of the tree). The paper experimentally compares this method against local explanations and a clustering method in terms of its ability to approximate f and give useful explanations to actual users in a user study. Additionally, a qualitative case study is enclosed to validate explanations with a domain expert.

Strengths: The papers strengths are the user study which gives a good indication of practical utility of the explanations, at least for experienced users, and that the method nicely extends established notions of explainability on terms of linear models, which should make it interesting for a wide range of NeurIPS community members. Further, the proposed approach is well-justified in optimization theory.

Weaknesses: The most severe shortcoming of the paper is in terms of justifying key parts of the approach, I believe. In particular: * What is the function g? The paper introduces this as a coordinate-wise mapping from the input space into p-dimensional Euclidean space - but it is unclear what 'coordinate-wise' relates to and the paper - as far as I see - does never provide an example what g could be. The experiments use the identity map for g, I believe. In line with the LIME paper, g could be a mapping into a space of features with explanatory value to the domain, but this is not stated, either. * While the paper does justify the need for multi-level explanations, it is less clear that the multiple levels should take the shape of a tree and that a clustering of examples is a reasonable choice to structure the tree (one could, e.g. also think about using coarser and coarser features). * It is not obvious that a graph between training examples is a useful or natural way to inject domain knowledge. Indeed, in the experiments the graph is only used in this way in the qualitative study, whereas in all other cases the graph is build based on the output of the original model f, i.e. instances x and y get connected if f(x) and f(y) is similar. This ad-hoc choice of graph is not clear to me, either. In my mind, it would be much more natural for the approach to connect points that are similar in the input space, or in the space of g, which corresponds to the idea of clustering. Another severe weakness is that I cannot see why the proposed approach is so computationally intensive. From the complexity analysis in the appendix I gather that each optimization iteration is quadratic in the number of samples, which may become cubic if a linear number of iterations is required, but it seems counterintuitive for me that this complexity should require hours of compute on such strong machines as used here. This seems to me like an obstacle to practical applicability that would warrant more discussion in the paper. This runtime is also relevant because it probably prohibited a crossvalidation of the offline experiments, which would be needed to get a sense whether the differences reported in table 1 are relevant or just a result of the specific train/test split. Accordingly, I would strongly recommend to go deeper into the practical efficiency issues of the approach and speed it up to enable a more thorough analysis. Finally, the experimental evaluation could include a-priori interpretable models as another baseline. In particular, it would be valuable to see whether a direct linear model performs much worse on this data compared to random forest or MLP. If not, this would be much faster to train and explain.

Correctness: I find the theoretical arguments for convergence convincing and I believe that the proposed approach is well-founded in optimization theory. Regarding experimental methodology, I am less convinced. For the first experiment, I believe that a crossvalidation is necessary to convincingly judge the algorithm's performance. For the user study, significance testing and effect sizes would be useful to underline the main result. It would also be helpful to report whether experimenters and participants were blind regarding the method they evaluated.

Clarity: As mentioned above, I believe that there is room for improvement with regards to clarity. In particular, clarifying the following points may be helpful: * The introduction could already introduce the notion of 'explanation' that is pursued in this paper, in particular a linear approximation of a model. This would also help to elucidate Figure 1; as it stands, the bar plots in Figure 1 are not quite clear. * page 3: the 'co-ordinate wise map' g should be explained. I believe this needs to be a map into a space of human-readable features, but I might be wrong. An example would be helpful, too. In the experimental section, it should also be made explicit which g was used. * page 4: A global explanation for the entire tree would only result if W corresponds to a connected graph, I believe (since for high value of beta, w_{i, j} > 0 enforces equality of \theta_i and \theta_j and, by the transitive property, this would make all explanations equal). But if the graph is disconnected, an infinite beta would still yield several clusters (one per connected component). This could be made explicit. It would also be useful to give an example here how the graph can be constructed (e.g. with the heuristic mentioned in the experimental section). * I believe it would improve clarity to move Algorithm 1 to page 5. * Below Equation (9), E_\beta and E_k should be described. * Giving an indication of the computational complexity in the main paper would be very helpful. The derivation can remain in the supplement, but the final complexity should be stated. * The reasoning behind the graph construction heuristic in the experiments is not sufficiently clear to me; in principle, there could be very different explanations that give rise to a classifier's output at a certain confidence level. Wouldn't it be more intuitive to, rather, consider similarity in the input space, e.g. by constructing an agglomorative clustering of the data or something similar?

Relation to Prior Work: My impression is that the paper describes related work sufficiently and relates its own contribution appropriately with prior work. I see no issue in this regard.

Reproducibility: Yes

Additional Feedback: I note that I did not reproduce the main results, but the documentation given in the supplement seems sufficient to do so. I believe that the broader impact statement is thoughtful and covers both potential benefits and risks well. This is, I think, a good example of how such a statement should be done. typos & style: * page 2: 'Based on expert feedback these intermediate explanations although explain the same type of pump have different manufacturers resulting in some noticeable difference in behaviors' I could not parse this sentence; perhaps there is a grammar mistake

[Author Response · NeurIPS 2020]

We thank the reviewers for their thoughtful comments. We will incorporate the clarifications provided here in the final
paper, add suggested references, and address minor comments such as typos, moving contents between the main paper
and supplement (since NeurIPS typically allows an extra page). Citations and line numbers refer back to the paper.
**Conceptual Clarifications:** [R1] *Meaning of level?* Leaves are level 1, increasing up to the root. [R1] *Will explanations*
*lose fidelity as we move up the tree?* Not always, since closeby train points maybe more realistic than perturbed (local)
neighborhoods (see Supp. Figures 3c, 3e, 4b and 4e). [R1] *Weight generation in absence of prior knowledge.* By sorting
the labels $f(x_i)$ and setting $w_{ij}$ to 1 whenever $f(x_i)$ and $f(x_j)$ are right next to each other in the sorted list (see lines
218-221). This simple prior (*path graph*) enforces that weights for similar observations must be similar. [R1] *Two*
*step: p, F, and tilda on parameters.* $F \rightarrow$ Frobenius norm, $p \rightarrow$ size of each parameter vector (line 93). Tildas on the
parameters differentiate them from MAME solutions. [R1] *SP-LIME relevant because it attempts to provide global*
*explanations by outputting a set of local explanations?* Yes, for comparisons, we control the size of this set to generate
same number of explanations as our method for any level. [R2, R3, R4] *Specification of $g()$.* For LIME and our methods,
$g()$ is an identity map for numeric and one-hot encoding for categorical features. However, it can be any non-linearity
applied on inputs such as squaring of features or multiplying features together to create meaningful interaction terms.
[R2] *Why does optimization in Eq. 1 or 2 not involve g?* $g()$ is pre-specified and not learned, hence the optimizations do
not involve them. [R2] *Regarding $\alpha_i$s:* $\alpha_i$ controls the sparsity of each local (leaf-level) explanation, and are tuned only
once to achieve the required individual sparsity for each example. It is not part of main optimization, and does not cause
overhead. [R2] *Why not L1 penalty in 3rd term of eq. 1?* L2 is most commonly used in convex clustering literature [21].
We tried L1, but L2 performed better. [R2] *Choice of $\mathcal{N}_i$:* $\mathcal{N}_i$ is obtained by randomly perturbing $x_i$, $m$ times (See
lines 98-99 and 215). [R2] *Relation to MAPLE:* We will add more discussion about MAPLE in the paper, however,
Two-Step and SP-LIME are closer competitors to our method. MAPLE creates a random forest which itself one might
argue cannot be directly interpreted. More importantly though, it does not provide group level explanations as the RF
model is learned over the entire dataset. [R3] *Requirement of prior knowledge:* A reasonable prior is important but
MAME outperforms other methods when this was chosen in a simple data-driven way. Only in one part of expert study
did we incorporate domain knowledge. [R4] *Is example-based grouping a good choice for multi-level?* To explain
groups of points in a principled way was our main motivation (like [16, 17]). Pursuing other flavors of multi-level (as
in [18]) will be a separate effort. [R4] *Prior knowledge via graph on $x$ or $f$?* Graphs based on $f(.)$ use the intuition
that explanations should be similar if the predictions are similar. It is unclear what the right metric for graphs with
(especially a high dimensional) $x$ would be.[R4] *Disconnected prior graph:* This will yield multiple clusters as $\beta \rightarrow \infty$.
**Importance of Results:**[R4] *Significance of user study:* MAME is statistically significantly better than Two Step and
SP-LIME in probability estimate and cluster assignment tasks. Will add ANOVA (for repayment probability MSE,
F(2, 27)=9.48, p < .0001, effect size=0.41) and post-hoc comparison results. [R1, R2, R3] *Regarding dataset results:*
With infidelity measure, MAME is better than Two-step in 7/10 cases and overall best in feature importance rank
correlation (4/5 cases). With generalized fidelity measure (Supp.), on an average (over all datasets) MAME improves
upon SP-LIME and Two Step by 18% and 16% for RF, and 32% and 18% for MLP. We also ran 5-fold CV for the
datasets (except ATIS since test partition is pre-defined), averaging over all of them MAME improves upon SP-LIME
and Two Step by 31% and 9% for RF, and 118% and 0% for MLP. For MLP, MAME was worse than Two Step only on
1 dataset which hurt the gains. [R4] *Runtime:* We have an efficient implementation in Julia. MAME took $\sim$ 4 hrs on
ATIS, where Two-Step took $\sim$ 6 hrs and SP-LIME took $\sim$ 4 days. For other smaller datasets, MAME took < 10 mins.
**User study with credit dataset:** [R1] *Regarding clusters and guessing them:* The clusters are not the same, but the
number is to keep it fair. How well the participants guess the cluster membership shows how homogeneous our clusters
are in terms of feature contributions. It also tests whether our explanations are simulatable by the user which is an
important metric to judge efficacy of explanations (Lipton 2016). We included the explanation figure for SP-LIME in
Figure 8 (supp.) [R1, R3] *Why can users guess outputs better with MAME?* Our method was i) selective, ii) created
homogeneous clusters (similar important feature values in each cluster), and iii) was still accurate in terms of the
prediction in each cluster. [R4] *Were participants blind to the methods?* Yes, we will clarify this.
**Expert study with Oil & Gas:** [R1] *Why were there 4 clusters at level 380?* The algorithm successively merges
clusters as we go up the tree, and hence resulted in 4 clusters at level 380. [R1] *Semantic relevance of MAME clusters -*
*reason?* MAME is able to ingest prior knowledge effectively and forms clusters whose explanation models are thus
not only more intuitive to the expert but also have high fidelity to the black-box model. Two Step does not explicitly
control for fidelity which results in less semantically relevant clusters. [R3] *Importance of capturing prior knowledge in*
*explanations:* Behavior of pumps vary widely according to manufacturer, which is why it is important to capture this
prior. Remaining useful life estimation of pumps is done separately for each manufacturer.
**Quantitative evaluations with public data:** [R1] *Comparisons to Pedreschi et al. (2019).* This paper just outlines
a high level approach, and does not actually propose an algorithm, so nothing specific to compare against. [R1]
*Controlling the amount explained?* In Table 1 (infidelity), the average is computed for all test examples over all levels
of the tree/representative explanations to ensure fairness. [R1] *Intuition for infidelity in 4.2?* This measure captures how
well our explanation can track changes in black-box prediction when the input goes from a chosen *null* to actual value.
Limitation is that this depends on the choice of *null*. See Supp. for another measure (*generalized fidelity*).

[Meta-Review · NeurIPS 2020]

Overall, the reviewers like the contributions of the paper and feel it will be a valuable contribution. However, they strongly urge that the authors describe the limitations of the work, in particular, the scalability issues that arise from the need to tune the alpha-i for each instance, and clarify the concern with using pre-defined graphs (and how it can significantly bias the results). These revisions will make the paper substantially stronger.